# The Evolving Role of Marked Lymph Node Biopsy (MLNB) and Targeted Axillary Dissection (TAD) after Neoadjuvant Chemotherapy (NACT) for Node-Positive Breast Cancer: Systematic Review and Pooled Analysis

**DOI:** 10.3390/cancers13071539

**Published:** 2021-03-26

**Authors:** Parinita K. Swarnkar, Salim Tayeh, Michael J. Michell, Kefah Mokbel

**Affiliations:** The London Breast Institute, The Princess Grace Hospital, London W1U 5NY, UK; Salim.tayeh@nhs.net (S.T.); mikemichell@aol.com (M.J.M.)

**Keywords:** breast cancer, node positive, systematic review, targeted axillary dissection

## Abstract

**Simple Summary:**

The 5-year survival rate for patients with breast cancer, in whom disease has spread to local lymph nodes, is 85%. However, many live with the complications of surgery to remove the lymph nodes in the armpit thus impacting their quality of life. In recent years, new approaches have been developed to minimise surgery and reduce complications. The aim of this systematic review was to assess the feasibility and accuracy of two minimally invasive surgical procedures, Marked Lymph Node Biopsy and Targeted Axillary Dissection as an alternative to complete removal of the axillary lymph nodes after upfront chemotherapy in patients in whom cancer spread to the regional lymph nodes. Our findings confirm that these procedures can safely replace more radical surgery in women who have responded well to upfront drug treatment. Therefore, although further research to determine long-term outcomes is required, this review concludes that it is reasonable to offer such patients the option of less invasive surgery thus avoiding over treatment and enhancing quality of life.

**Abstract:**

Targeted axillary dissection (TAD) is a new axillary staging technique that consists of the surgical removal of biopsy-proven positive axillary nodes, which are marked (marked lymph node biopsy (MLNB)) prior to neoadjuvant chemotherapy (NACT) in addition to the sentinel lymph node biopsy (SLNB). In a meta-analysis of more than 3000 patients, we previously reported a false-negative rate (FNR) of 13% using the SLNB alone in this setting. The aim of this systematic review and pooled analysis is to determine the FNR of MLNB alone and TAD (MLNB plus SLNB) compared with the gold standard of complete axillary lymph node dissection (cALND). The PubMed, Cochrane and Google Scholar databases were searched using MeSH-relevant terms and free words. A total of 9 studies of 366 patients that met the inclusion criteria evaluating the FNR of MLNB alone were included in the pooled analysis, yielding a pooled FNR of 6.28% (95% CI: 3.98–9.43). In 13 studies spanning 521 patients, the addition of SLNB to MLNB (TAD) was associated with a FNR of 5.18% (95% CI: 3.41–7.54), which was not significantly different from that of MLNB alone (*p* = 0.48). Data regarding the oncological safety of this approach were lacking. In a separate analysis of all published studies reporting successful identification and surgical retrieval of the MLN, we calculated a pooled success rate of 90.0% (95% CI: 85.1–95.1). The present pooled analysis demonstrates that the FNR associated with MLNB alone or combined with SLNB is acceptably low and both approaches are highly accurate in staging the axilla in patients with node-positive breast cancer after NACT. The SLNB adds minimal new information and therefore can be safely omitted from TAD. Further research to confirm the oncological safety of this de-escalation approach of axillary surgery is required. MLNB alone and TAD are associated with acceptably low FNRs and represent valid alternatives to cALND in patients with node-positive breast cancer after excellent response to NACT.

## 1. Introduction

Due to the significant associated morbidity, complete axillary lymph node dissection (ALND) has been largely replaced by the less-invasive sentinel lymph node biopsy (SLNB) as the gold standard for regional axillary staging in clinically node-negative breast cancer patients undergoing upfront surgery [1]. A recent meta-analysis of 16 studies spanning 1500 patients confirmed that the SLNB was reliable in staging the axilla in patients with cN0 breast cancer after neoadjuvant chemotherapy (NACT) with an overall identification rate of 96% and a false-negative rate (FNR) of 5.9%. The latter is well below the target limit of 10% [2,3].

Furthermore, retrospective studies provided evidence that the SLNB is oncologically safe in this setting [4,5]. In a retrospective analysis, patients with cN0 and T1–T3 disease underwent either SLNB after NACT (*n* = 575) or first-line surgery (*n* = 3171). Results demonstrated axillary recurrence of 1.2% in the NACT group, with no difference in disease-free or overall survival between both groups [6]. Retrospective studies provided evidence that the SLNB is oncologically safe in this setting [7].

Women diagnosed with biopsy-proven node-positive breast cancer are usually considered for NACT, which has been shown to be beneficial in reducing tumor burden and increasing the success rate of breast-conserving therapy (BCS) [8]. Furthermore, this approach provides critical information regarding the tumor responsiveness to systemic therapy and the potential need for a new class of drugs in the adjuvant setting when pathological complete response (pCR) is not attained [2]. Complete ALND has been the gold standard surgical management of the axilla in patients with node-positive breast cancer receiving NACT. Two recent trials (AMAROS and ACOSOG Z0011) demonstrated that omission of ALND in patients with a positive SLNB does not compromise overall survival [2]. This has stimulated interest in extending de-escalation of axillary surgery to patients with cN1 breast cancer that responds well to NACT [3].

However, in biopsy-proven node-positive patients undergoing NACT, SLNB studies have reported inconsistent false-negative and identification rates [9]. Our recent meta-analysis of approximately more than 3000 patients with node-positive breast cancer reported a FNR of 13% after NACT, which is above the threshold target of 10% [10]. The studies included in our meta-analysis were heterogeneous, retrospective and nonstandardized in nature. The variation in FNRs in this setting has been attributed to anatomical changes resulting in aberrant lymphatic drainage, consequential NACT-associated fibrosis, fat necrosis and/or granulation tissue formation or the tumor itself [11].

Targeted axillary dissection (TAD) is a new axillary staging technique whereby the lymph node-positive for metastatic disease at initial diagnosis is marked using different methods such carbon tattooing, radioiodine, metallic clips, ferromagnetic seeds, etc., prior to NACT so that this marked lymph node (MLN) can be removed during breast cancer surgery [12]. Intuitively the MLN biopsy (MLNB) should reflect the status of residual axillary disease more accurately than the SLNB alone in this setting. The MLN is usually identified, biopsied and marked using ultrasonography of the axilla for guidance. If the MLN cannot be identified or remains positive for metastatic disease after NACT, ALND is usually carried out.

There have been numerous studies, with varying results, that have investigated the role of MLNB and TAD in this context [13,14,15,16,17,18,19,20,21,22,23,24,25,26,27,28,29,30,31]. The aim of this study was to evaluate the FNR of MLNB alone and combined with SLNB (TAD) by pooling the data from the relevant studies.

## 2. Materials and Methods

### 2.1. Data Sources and Searches

The study was approved by the multidisciplinary breast cancer board of the London Breast Institute.

Comprehensive searches of PubMed, Google Scholar and Cochrane Library databases were performed to identify and extract publications and records relevant to this study. The search strategy of the databases included key words such as: targeted axillary dissection, axillary lymph node clearance, neoadjuvant chemotherapy, node-positive breast cancer and false-negative rate. A further advanced search was conducted using the combination of words or phrases and abbreviations, using Boolean operators (“AND”, “OR”, “NOT”). The PubMed and Cochrane Library databases were searched on two separate occasions: 8 November 2020, and 17 November 2020. A final literature search was conducted on 23 January 2021. Furthermore, bibliographies from the included reviews and articles were manually screened for additional relevant publications. This ensured relevant and poorly indexed papers were not overlooked and that applicable studies, that did not come under the remit of the aforementioned search terms, were included. 

### 2.2. Inclusion and Exclusion Criteria

This analysis included both retrospective and prospective cohort studies. All publications were required to have summarized findings in the abstract regarding the effect of MLNB alone or combined with SLNB on the FNR in post-NACT patients with biopsy-proven node-positive breast cancer who underwent ALND. Of the studies that met these requirements, the full texts (where available) were reviewed, and the following raw data were required to be included:Total number of patients undergoing MLNB;Total number of patients undergoing TAD;Number of false-negative events/patients with false-negative results;FNR (%).

Abstracts not evaluating the FNR of MLNB or TAD in post-NACT biopsy-proven node-positive breast cancer patients were excluded. In addition, studies were excluded that were not peer-reviewed and in which the data were unclear or unavailable. This analysis excluded studies that were published in languages other than English and those with nonhuman subjects. Publications comparing MLNB or TAD to other axillary staging techniques were included in this analysis, and data regarding these methods were ignored for the purposes of our calculations, except where relevant to the smaller pooled analysis. In addition, both full texts and abstracts were included in this review.

### 2.3. Data Management

Data extracted from eligible studies include the first author, FNR, absolute false-negative number and total patient number. We extracted and combined the FNRs of the included studies to calculate the overall rate of false negatives for MLNB and TAD from the datasets. By combining data sets from all included studies, the mean values were calculated to provide the overall FNR of MLNB (± SLNB) in post-NACT for biopsy-proven node-positive breast cancer.

Furthermore, the confidence intervals were calculated using SciStat^®^ and testing for statistical significance (5%) in the difference between FNRs was calculated using the Chi-squared test.

## 3. Results

### 3.1. Literature Search Results and Characteristics of the Included Studies

In the preliminary search, a total of 174 records were identified (157 from Google Scholar; 16 from PubMed; 1 from the Cochrane Library). Following the removal of duplicates and the addition of records identified from other sources, 149 publications were reviewed for inclusion. Once these studies were screened for eligibility, 138 were immediately excluded. The full texts (where available) were examined for the remaining 11 studies. Manual searching yielded eight relevant abstracts and/or full text from the respective bibliographies. 

### 3.2. Results of Pooled Analysis

Through analysis of the 9 included studies (Table 1), a total of 366 post-NACT patients with biopsy-proven node-positive breast cancer underwent MLNB in addition to cALND [13,14,15,16,17,18,19,20,31]. Of these, 23 false-negative results were recorded yielding a FNR of 6.28% (95% CI: 0.03984–0.09429). In 12 studies (Table 2) spanning 521 patients that evaluated the addition of MLNB to SLNB (TAD), we calculated an overall FNR of 5.18% (95% CI: 0.0341–0.0754) [12,13,15,17,19,21,22,23,24,25,26,27,31].

Statistical analysis showed that the difference in FNR between MLNB and TAD was not statistically significant (*p* = 0.484, Chi-squared statistic = 0.489).

Analysis of studies (Table 3) [10,11,12,13,14,15,16,17,18,19,20,22,23,25,26,27,28] reporting the technical success of localizing and retrieving the marked lymph node revealed a summative successful retrieval rate of 90.0% (95% CI: 0.8515–0.9505).

Our search did not reveal any prospective studies reporting long-term survival data in patients with ypN0 undergoing MLNB or TAD.

## 4. Discussion

We have systematically reviewed the feasibility of MLNB alone and in combination with SLNB in patients with biopsy-proven node-positive breast cancer who received NACT and quantified an overall FNR of 6.28% for MLNB alone and 5.16% for TAD. These FNRs are significantly below our reported FNR for SLNB alone of 13% in the same setting [10]. Furthermore, the FNRs observed in our pooled analysis are below the accepted target of 10% for clinically proven node-negative breast cancer [2,3]. This pooled analysis has confirmed that MLNB is an accurate technique in axillary staging after systemic therapy and is associated with a high technical success rate and an acceptably low FNR. Although the addition of SLNB to the MLNB was found to be associated with a lower FNR, the difference of 1.12% was not statistically significant. On the contrary, most studies demonstrated that the added evaluation of MLNB significantly decreased the FNR of SLNB alone [13,21,27].

Therefore the SLNB can be safely omitted in the context of a successful MLNB, thus reducing costs and additional complications. Optimizing the FNR in patients receiving NACT for a biopsy-proven node-positive breast is important because understaging the residual axillary disease can potentially result in adjuvant systemic therapy undertreatment and compromise of oncological outcome in patients particularly with HER2-positive and triple-negative breast cancer (TNBC) where residual disease is used to guide the use of further adjuvant systemic therapy, such as capecitabine for TNBC and TD-M1 for HER2-positive disease [2].

### 4.1. Clips

The initial techniques of marking the target lymph node included the deployment of clips made of various materials such as stainless steel, titanium or polyglycolic acid [32] and carbon or black ink tattooing [33]. The majority of the studies included in the current analysis were based on TAD, implementing the use of clips deployed within the pathological lymph node in a procedure following initial percutaneous ultrasonography-guided biopsy. It is recommended that the marker is deployed at the time of biopsy to obviate the need for a second procedure [21] Tattooed target lymph nodes can be visualized directly during surgery; however, the marker clips require a second localization procedure prior to surgery, and accurate identification with ultrasonography can be challenging when the pathological lymph nodes revert to normal after NACT. Displacement of the marker clip into the surrounding perinodal fat and fibrous tissue (secondary to node shrinkage post-NACT) can be also a contributing factor to difficulty localization by ultrasonography [15]. HydroMark (Devicor Medical Products), which consists of a metal clip made of titanium or stainless steel embedded in a hydrogel made primarily of collagen, has the highest degree of visibility on ultrasonography up to 12 months; however, some authors cautioned that ultrasound visibility after six months can be significantly reduced due to collagen resorption [27]. Displacement of the wire used for the localization of the clip-containing target lymph node represents another limitation to the use of clips [15]. These limitations stimulated interest in developing new methods that do not require a second localization procedure and wire insertion [25,29,34,35].

Ten studies included in this review evaluated the implementation of clipped nodes, and the range of the reported FNR is from 0.0% to 7.3% [12,13,15,17,18,21,22,23,25,27,31]. These studies varied in protocol and focus (MLNB alone or TAD), but the majority concluded that further studies, with larger sample sizes and longer follow-up, would be required to ensure oncological safety and for clinical application [12,17,18,25,31]. 

Of the FNRs, 0.0% were concluded from studies evaluating TLNB and TAD separately and with small sample sizes (collectively 47 patients) [15,31]. Flores et al. concluded that, although a larger sample size would be required for validation, SLNB did not provide additional diagnostic value and could be omitted in their cohort of patients. Opposingly, Hartmann et al. deemed their procedure inappropriate for clinical use given clip-displacement and identification issues (27.3% not reliably confirmed by removal). 

The highest FNRs of this subgroup (7.3% and 7.1%) were both from pilot studies. Direct comparisons, and therefore drawing generalizable conclusions, are difficult given the notable differences in patient numbers (96 and 14, respectively) [12,18]. In addition, the two studies varied largely in the protocol applied. Mittendorf et al. elected to clip the single “most suspicious” node, whereas, Lim et al. clipped all nodes of abnormal appearance [12,18]. Both studies sonographically determined abnormal nodes. This raises the possibility of limitations in the identification of abnormal nodes by ultrasound. Lim et al. recommend a maximum of three abnormal nodes as a cut-off for TAD. This is agreed upon by Cabioglu et al. (reasons not given) [22]. 

Only two studies in this subgroup included more than 100 patients [13,21]. Caudle et al. reported a FNR of 10.1% in SLND, 4.2% in MLNB and 2.0% in those undergoing the TAD procedure, thus demonstrating a decrease in FNR associated with TLNB and a further decrease associated when combined with SLND. The study concluded that this procedure improves axillary staging accuracy and enables the sparing of ALND in a relevant number of patients. The ILINA trial and Kuemmel et al. reported similar FNRs (4.1% and 3.9%, respectively) [17,22]. All three studies implemented dual-localization for the identification of the clip. One main difference between the studies was identification: Siso et al. used intraoperative ultrasound (IOUS), Kuemmel et al. used wire-guided technology and Caudle et al. involved the placement of an Iodine-125 seed. Both Kuemmel et al. and Siso et al. ensured immunohistochemical (IHC) staining in unclear specimens to reduce undertreatment, whereas Caudle et al. recommended this as an improvement to their own protocol to reduce the FNR [13,17,27].

Boughey et al. concluded that the FNR was higher for cN1 patients undergoing resection of multiple (≥2 nodes) when the clipped node was found within the ALND specimen (19.0%) than in the SLN specimen (6.8%) [21]. This result is comparable to an FNR of 16.7% [22] when clipped nodes could not be identified within the SLN specimen. Kuemmel et al. also evaluated the FNR in patients with ≥2 nodes resected but were unable to determine statistical significance given underpowered subgroup analyses [17]. 

Localization issues also surround the use of clips in TAD. Three studies implemented wire-guided localization techniques. Hartmann et al. and Flores-Funes et al. (both FNR 0.0%) comment on the difficulty of use and recommend obviating a second invasive procedure [15,31]. Siso et al. (FNR 2.9%) exclusively used IOUS for localization, and Kuemmel et al. (FNR 3.9%) used IOUS in cases where initial wire-guidance was unsuccessful [17,27]. Martinez et al. reported the use of MagSeed. This enables the benefits of the Iodine-125 seed localization (used by Caudle et al.) without radiation and regulatory restrictions [13,25], or a significant increase in FNR (5.9% and 4.2%, respectively). 

### 4.2. MARI

Marking axillary lymph nodes with radioactive iodine (^125^I) (MARI) seeds that can be localized using a gamma probe was used in three publications included in this analysis [14,16,19]. This technique stems from the success in localizing residual breast disease having implanted radioactive iodine seeds prior to NACT in the center of the primary tumor [19,36]. The MARI technique is straightforward and easy to learn and perform by surgeons experienced in SLNB dissection. Furthermore, it obviates the need for wire insertion. From a surgical perspective, another advantage is the long half-life of the iodine seed, around 60 days, allowing adequate time for NACT and bypassing scheduling conflicts associated with the much shorter half-life (around 8 h) of radiocolloids used in dual-tracer SLNB. Radioactive iodine seeds are associated with a decreased displacement risk in the time between insertion and surgery and, therefore, a decreased risk of injury to vascular structures in the surrounding area [16,37]. 

However, the use of radioactive materials is complicated by complex regulatory requirements. Moreover, the time that the seed can stay within the human body is limited to 5–7 days in certain jurisdictions, thus prohibiting deployment of the seed at the time of biopsy prior to NACT. Prior to insertion, fine-needle aspiration biopsy (FNAB) is required to determine the intended lymph node for seed insertion. This additional procedure mandates documentation to ensure localization in a disease-free negative node is avoided [14]. Furthermore, a single node (typically the largest) is selected for FNAB and seed insertion. It may be beneficial to mark multiple abnormal nodes (provided the radioactivity levels retain patient-safety). A potential complication with this method concerns the theoretical sterilization of the selected lymph node due to the minimal radioactivity of the seed [14].

The MARI subgroup consists of three studies, and the FNR range is 0.0% to 15.6% [14,16,19]. The Iodine-125 seed placement duration (median: 17 to 18 weeks) and protocol were similar across the three studies. The essential differences were sample sizes and initial axillary staging methods: ultrasonography [14,19] or [18F]FDG PET–CT) [16]. Very few groups have evaluated the FNR in the MARI technique, and the included studies recommend further testing in larger cohorts and longer-term follow-ups. The Koolen et al. study further subcategorized patients according to the number of positive lymph nodes prior to NACT. The reported FNR was higher for patients with ≥4 nodes (20.0%) than 1–3 nodes (13.6%) [16]. In line with these results, and as a general consensus, patients with ≥3 nodes are considered postoperative locoregional radiotherapy (RT) candidates [38]. Despite the FNR, the study concludes that 74% of patients avoided ALND, with a limited risk of undertreatment. Furthermore, this study acknowledges the cost of [18F]FDG PET–CT and regulatory issues involved with radioactive seed use. The paper discusses the alternative role of Iodine-125 seeds in localization. This method has been used by Caudle et al. (included in this review) and reported a low FNR. Donker et al. demonstrated method feasibility by a reduced FNR (7.1%) in 70 patients [14]. This could provide justification for omitting postoperative RT in patients with MARI-confirmed node-negative disease. Straver et al. reported the lowest FNR (0.0%); however, they also had the smallest sample size (15 patients) [19]. The authors recommend confirmation of the findings by a larger study. 

### 4.3. Carbon Tattooing

Carbon tattooing of the metastatic node was implemented in three publications [20,24,26] included in this analysis. All tattooing was performed under ultrasound guidance. Reported advantages of this technique include the ease of intraoperative identification and no requirement of an invasive localization procedure, thus reducing burden for the patient and avoiding the use of radioactive materials [20,26]. A previous study described a “dual-localization” technique whereby a metastatic lymph node is marked with a clip (prior to NACT) and tattooed with activated charcoal (after NACT). This was performed to circumvent a second localization procedure and unavailability of radioactive seeds in many countries [39]. Furthermore, the black ink used by Park et al. was found to be detectable for up to 197 days post-tattooing, thereby allowing appropriate time for NACT [26]. Potential tattoo pigment migration to other lymph nodes in addition to the need for a wider surgical dissection to visualize the tattooed nodes represent limitations to using this technique.

Three studies are included in this subgroup analyses, and the FNR range is 0.0% to 7.0% [20,24,26]. All three studies concluded this was a safe technique but that further, and larger, studies would be necessary to determine clinical implications and long-term oncological safety. 

Gatek et al. had the smallest sample size (eight) and the lowest FNR. No complications were recorded, and the study included only patients with ductal carcinomas [24]. The study by Park et al. recorded a 100% node-detection rate and implemented mandatory H&E staining, with or without IHC [26]. Furthermore, the authors emphasized the benefits of this procedure by accurate localization via needle-point injection track from the ALN to the axilla-skin surface: no preoperative localization procedure for the patient and removes technical mistakes associated with anatomical variants (e.g.: Langer’s axillary arch). This study implemented the use of dual-localization for the SLNs (unlike the other two studies in this subcategory). Spautz et al. outlined tattooing as the most suspicious node in patients where multiple abnormal nodes were identified via ultrasound [20]. They reported a higher detection rate with carbon tattooing (98.3%) than patent blue V (PBV) dyed SLNs (91.0%), higher than that reported by Caudle et al. (95.8%) when using an Iodine-125 seed [13]. The authors list the lack of dual-tracer localization as a limitation but make note that this is not widely available and therefore ensures this protocol is reproducible [20]. It is possible that the implementation of dual-tracer technology and IHC staining may have reduced the FNR (7.0%) of this study. Spautz et al. importantly emphasize that carbon tattooing serves to enhance SLNB rather than replacing it altogether. 

### 4.4. Localization

The limitations discussed above have inspired the evolution of novel radiation-free wireless technologies have emerged including magnetic seeds [25] (Magseed^®^; Endomagnetics Inc., Cambridge, UK), infrared reflectors [30] (Savi Scout; Merit Medical Inc., Aliso Viejo, CA, USA) and radiofrequency identification [28] (RFID) tags (LOCalizer; Hologic, Santa Carla, CA, USA). All three radiation-free wireless methods allow localization to occur before the day of surgical excision thus reducing the need for scheduling coordination. Magseed utilizes a 5 mm paramagnetic seed which is deployed through a sterile 18-gauge needle. It can be detected from the skin surface using a handheld probe up to a reliable depth of 4 cm. The relatively large detection probe size and the need to remove all metal instruments from the surgical field when the probe is in use represent important limitations to Magseed in the context of MLNB and TAD [40].

Savi Scout involves the insertion of a 12 × 1.6 mm^2^ electromagnetic wave reflector into the target lymph node using a sterile 16-gauge introducer needle delivery system. The reflector can be detected by a radar up to 6 cm depth from the skin. Electrocautery should be used with caution when performing MLNB or TAD using the Savi Scout system. Unlike Magseed and RFID tags, the reflector of Savi Scout does not generate significant MRI artifacts, and this is important if MRI is used to monitor response to therapy (Figure 1 and Figure 2) [35]. The radar reflection localization (RRL) of Savi Scout enhances the identification of the reflector by an audible sound and digital display of distance from the probe. Sun et al. reported recovery of all reflectors and no complications [30]. 

The LOCalizer utilizes a 10 × 2 mm^2^ tag with a glass casing that is deployed using a 12-gauge introducer needle. The detection probe is both user friendly and site specific. A unique five-digit number associated with each RFID tag enables both site-specific and user-friendly identification and this is of particular benefit in patients with multiple tag-deployment sites [28].

However, the size of the tag and the width of the introduced needle represent limitations particularly in patients with small pathological logical lymph nodes. These novel wire-free radiation-free techniques were used only in a small number of patients included in this analysis. Further research is required to determine their clinical performance in larger series and establish which one of these technologies will achieve optimal marking and localization of pathological lymph nodes in addition to analysis of cost effectiveness [41]. 

We have observed a pooled successful localization and retrieval rate of the MLN of 90%, and this means that the MLN is not received in 10% of cases. In such cases, complete ALND should be performed to ensure accurate staging. If the MLNB is performed as part of TAD, then the SLNB can be considered as an alternative to ALND in patients with a complete radiological response provided that a minimum of three sentinel nodes are harvested using the dual-localization technique [2,21] because the FNR will be below 10% in this setting. In the ACOSOG Z1071 (Alliance) study, the MLN was found to be within the SLNB in 78% of cases when the dual-localization technique was used [21]. 

There is currently no consensus regarding the applicability of MLNB or TAD to patients with multiple pathological lymph nodes. Lim et al. raised concerns regarding differential axillary nodal response to NACT [18]. After chemotherapy, the patients in the study underwent removal of the clipped nodes using the Skin Mark clipped Axillary nodes Removal Technique (SMART) and ALND. The first clipped node predicted the axillary status with a false-negative rate of 7.1%. Adding to this another second clipped node, the false-negative rate was 0%.

The conservative National Comprehensive Cancer Network (NCCN) guidelines permit the use of TAD in patients who present with biopsy-proven node-positive disease if only one or two suspicious nodes are found on imaging, these positive nodes are not palpable clinically, and the other eligibility criteria from the Z0011 study are otherwise met [42]. 

Their main barrier to routine implementation of TAD and MLNB as part of standard clinical practice is the paucity of data regarding oncological safety. Although ALND may not be needed for patients with limited residual nodal burden and biologically favorable tumors, SLNB alone was reported to be inferior to ALND in patients with ypN1 disease following NACT in terms of five-year survival in a recent retrospective study [43]. Although there are currently no prospective studies reporting long-term survival data in patients with ypN0 undergoing MLNB or TAD, we have estimated that the probability of compromising overall survival (OS) or disease-free survival (DFS) would be approximately 1 in 2000 for a FNR of 10% and 1 in 10,000 for a FNR of 2% if ALND is omitted [2,3]. Therefore the benefit–risk balance would favor TAD to ALND in this patient population.

Further evidence regarding the oncological safety of axillary surgery de-escalation in patients rendering SLNB negative after NACT for cN1 breast cancer has been provided by data analysis of a large European study [44]. 

The main limitations of our study include the heterogeneous nature of studies included in the analysis and lack of standardized inclusion criteria, methods of marking and localization and definition of response to NACT and selection criteria for MLNB or TAD. Furthermore, pathological examination of the MLN was not standardized [17]. Caudle et al. suggested that the FNR of TAD could be lower if immunohistochemistry was to become a routine part of pathological evaluation [13]. Moreover, most studies had a small sample size (less than 100), and selection and publication bias could not be excluded. 

Ongoing prospective trials aim to provide important data regarding the optical technique and long-term oncological safety. Nijnatten and colleagues commenced a prospective multicenter validation study in 2017. Expected completion of the trial was in 2020, and the results are awaited [45]. The study aims to test the feasibility of MARI and SLNB. Another prospective multicenter trial, led by Henke et al., aims to publish data regarding DFS in patients who have undergone TAD and axillary radiotherapy and establish this procedure as a valid alternative to cALND [46]. 

## 5. Conclusions

The present pooled analysis demonstrates that MLNB and TAD are feasible with a high technical success rate and an acceptably low FNR in patients responding well to primary chemotherapy for node-positive breast cancer. Successful implementation of the technique requires careful multidisciplinary collaboration between breast radiologists, breast surgeons and breast pathologists. Further research to determine the optimal technique, standardize selection criteria and confirm oncological safety is required.

## Figures and Tables

**Figure 1 cancers-13-01539-f001:**
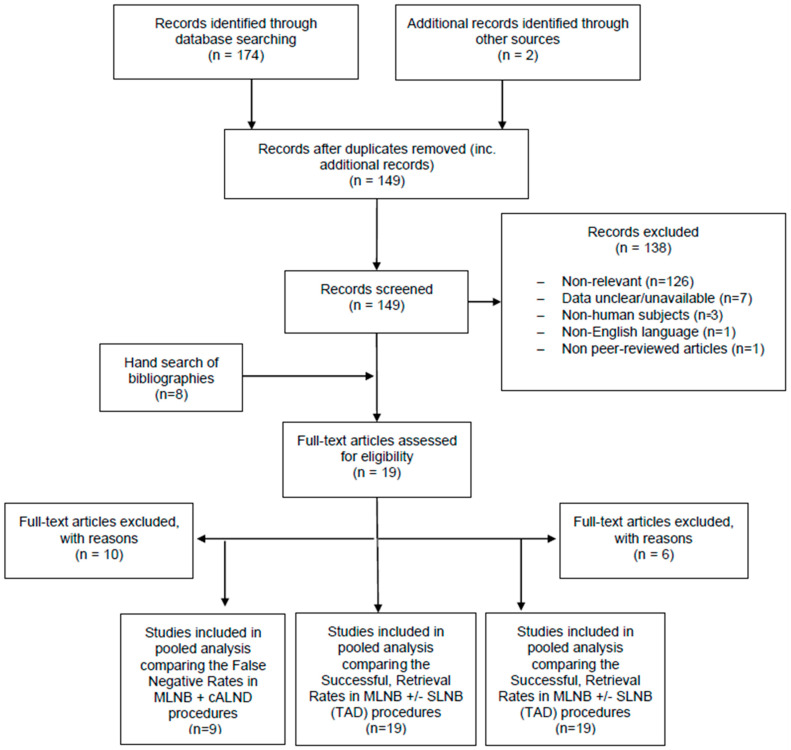
A PRISMA flow chart summarizing the results of data collection.

**Figure 2 cancers-13-01539-f002:**
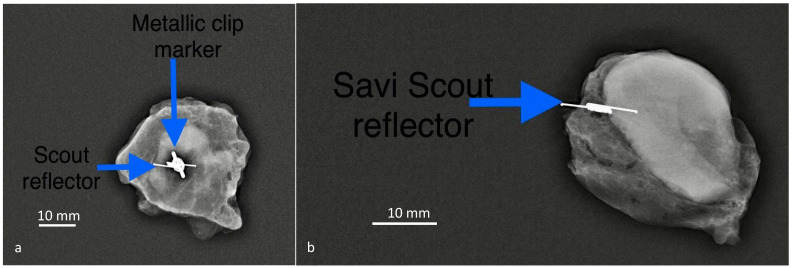
In (**a**), the patient had a metallic marker clip deployed within the pathological lymph node at the time of biopsy before neoadjuvant chemotherapy (NACT) and a second localization procedure using Savi Scout prior to surgery. In (right), the patient had the Savi Scout reflector at the time of biopsy prior to NACT thus avoiding a second procedure. There were no MRI artifacts related to the Savi Scout reflector in (**b**). The surgical procedure of identification and retrieval of the Savi Scout reflector took 15 min. Scale bar: 10 mm.

**Table 1 cancers-13-01539-t001:** Marked lymph node biopsy (MLNB) studies and false-negative rate (FNR).

Study	Year	Number of False Negatives	Total Patients	Method of Marking Targeted Lymph Node (TLN)	Localization Method
Caudle et al. [13]	2016	5	120	Metallic clip	Iodine-125 seed
Donker et al. [14]	2015	5	70	Radioiodine seed	Gamma probe
Flores-Funes et al. [31]	2019	0	23	Metallic clip	Wire guided
Hartmann et al. [15]	2018	0	3	Metallic clip	Wire guided
Koolen et al. [16]	2017	5	32	Radioiodine seed	Gamma probe
Kuemmel et al. [17]	2020	4	46	Metallic clip	Wire guided
Lim et al. [18]	2020	1	1	Metallic clip	NR
Spautz et al. [20]	2020	3	43	Carbon tattooing	N/A (visualized)
Straver et al. [19]	2010	0	15	Radioiodine seed	Gamma probe
Total		22	366		

NR—Not Reported. N/A—Not Applicable.

**Table 2 cancers-13-01539-t002:** Targeted axillary dissection (TAD) (MLNB + sentinel lymph node (SLN)) studies and FNR.

Study	Year	Number of False Negatives	Total Patients	Method of Marking Target Lymph Node (TLN)	Localization Method
Boughey et al. [21]	2017	7	107	Metallic clip	NR
Cabioglu et al. [22]	2018	1	24	Metallic clip	NR
Caudle et al. [13]	2016	1	74	Metallic clip	Iodine-125 seed
Coufal et al. [23]	2018	0	35	Metallic clip	Full abstract unavailable—Not determined
Flores-Funes et al. [31]	2019	0	23	Metallic clip	Wire guided
Gatek et al. [24]	2020	0	8	Carbon tattooing	N/A (visualized)
Hartmann et al. [15]	2018	0	3	Metallic clip	Wire guided
Kuemmel et al. [17]	2020	2	46	Metallic clip	Wire guided
Martinez et al. [25]	2020	1	17	Metallic clip	Magseed
Mittendorf et al. [12]	2014	7	96	Metallic clip	NR
Park S et al. [26]	2018	1	24	Carbon tattooing	N/A (visualized)
Siso et al. [27]	2018	1	24	Metallic clip	Intraoperative ultrasound (IOUS)
Straver et al. [19]	2010	0	15	Radioiodine seed	Gamma probe
Total		27	521		

NR—Not Reported. N/A—Not Applicable.

**Table 3 cancers-13-01539-t003:** Successful retrieval rate.

Study	Year	Number of Retrieved MLNs	Total Number of Marked Lymph Nodes	Method of Marking Target Lymph Node (TLN)	Localization Method
Boughey et al. [19]	2017	141	170	Metallic clip	NR
Cabioglu et al. [22]	2018	83	86	Metallic clip	NR
Caudle et al. [13]	2016	208	208	Metallic clip	Iodine-125 seed
Donker et al. [14]	2015	97	100	Radioiodine seed	Gamma probe
Flores-Funes et al. [31]	2019	22	23	Metallic clip	Wire guided
Gatek et al. [24]	2020	8	8	Carbon tattooing	N/A (visualized)
Hartmann et al. [15]	2018	17	24	Metallic clip	Wire guided
Koolen et al. [16]	2017	93	93	Radioiodine seed	Gamma probe
Kuemmel et al. [17]	2020	329	423	Metallic clip	Wire guided
Lim et al. [18]	2020	18	21	Metallic clip	NR
Lowes et al. [28]	2020	6	6	Radiofrequency identification (rfid) tags	RFID probe
Martinez et al. [25]	2020	29	30	Metallic clip	Magseed
Park S et al. [26]	2018	20	20	Carbon tattooing	N/A (visualized)
Siso et al. [27]	2018	35	35	Metallic clip	Intraoperative ultrasound
Spautz et al. [20]	2020	121	123	Carbon tattooing	N/A (visualized)
Straver et al. [19]	2010	15	15	Radioiodine seed	Gamma probe
Sun et al. [30]	2020	45	45	Metallic clip	Savi Scout System
Total		1287	1430		

NR—Not Reported. N/A—Not Applicable.

## Data Availability

The data supporting the results of this study are available in the Table 1 and Table 2 presented in the results section of this article.

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
