# Peer review of "The Evolving Role of Marked Lymph Node Biopsy (MLNB) and Targeted Axillary Dissection (TAD) after Neoadjuvant Chemotherapy (NACT) for Node-Positive Breast Cancer: Systematic Review and Pooled Analysis"

_cancers, 2021, doi:10.3390/cancers13071539_

Round 1
Reviewer 1 Report
In this manuscript, the authors determined the false-negative rate (FNR) of marked lymph node biopsy (MLNB) alone and targeted axillary dissection (TAD) (MLNB plus SLNB) compared with the gold standard of complete axillary lymph node dissection (cALND). They quantified an overall FNR of 6.28% for MLNB alone and 5.16% for TAD which was not significantly different from that of MLNB alone (p=0.48). Overall, the authors proposed that SLNB can be safely omitted from TAD. The paper is well written and in line with the Journal' aims.
I have some minor comments:
- Inclusion and Exclusion Criteria. It is not clear if only peer-reviewed articles were considered.
- Figure 1. I suggest minimizing the presence of the empty space. Panels must be presented in the portrait o landscape orientation in a single row. Please add a scale bar. -
- Line 194 and Line 232. Please add a full stop at the end of these sentences.
Author Response
Reviewer 1
Thank you for your comments, all of which have now been amended.
Please find below responses to each suggestion made.
1. Only peer-review articles were considered for inclusion - a sentence has been included within the manuscript and a flow diagram (to outline our inclusion/exclusion criteria) also outlines this
2. The figures have now been made smaller, and placed side by side in a landscape format. Additionally, a scale bar has been added (each representing 10mm).
3. Full stops have now been added at the specific lines as requested

Reviewer 2 Report
Generally, the aim of the study is interesting, based on an important topic for breast cancer management. However, there are some major and minor points to address to improve the quality of the manuscript. See the comments below:
Introduction – this section is acceptable, nevertheless a simplification in some sentences would provide a better comprehension by the reader.
- Line 35-39: The sentence is too long and confusing, please clarify.
- Line 39-40: The authors refer that there are studies providing evidence that the SLNB is oncologically safe; however, no bibliographic references were cited. Please add some references.
- Line 53-59: The sentence is too long and confusing, please clarify.
- Line 77-78: Please add some references that show the mentioned studies.
Materials and Methods
- Line 87: Why the search terms varied according to databases?
- I suggest the authors to add an illustrated flowchart of the excluded and selected studies for a better understanding of this relevant information.
Results
- Line 130: Please explain why the 138 studies were immediately excluded. Again, I strongly recommend the elaboration of a flowchart where the exclusion reasons are described.
- From the 147 studies reviewed for inclusion, 138 were excluded. Thus, the authors used 9 studies in this meta-analysis (table 1). Nevertheless, there are additional studies cited in table 2 and 3. Those studies are bibliography from the included reviews and articles that were manually screened (as referred in the line 94-95)? Clarify this and explain the reason for this type of procedure.
- First line of the table 3: reference is missing.
Discussion
Overall, this section could be improved. The results of the included studies should be more clearly debated.
Author Response
Reviewer 2
Thank you for your comments, these were very detailed and incredibly helpful in improving this paper.
Please find below the responses to each suggestion made.
Introduction
This section has been made clearer by clarifying specific points and reducing sentence lengths (where pointed out). Furthermore, we have added in the references at the points where requested.
Materials and Methods
The search terms did not vary per database - I do not know why this was in the original manuscript but has now been removed. All changes can be tracked via the document I have submitted below. As per your suggestion, a flow chart outlining our inclusion/exclusion criteria and process has been added. This also answers other suggested revisions made in other manuscript section.
Results
The flow chart outlines the criteria (as mentioned above), the reference has been added to table 3, and a sentence has been inserted in the Materials and Methods to explain our reasoning behind the manual screening of bibliographies.
Discussion
This section now has added paragraphs (see document attached), and the results have been debated in more detail. This includes both numerical comparisons (where appropriate) and reasons for similarities/differences in results.
Thank you very much for your comments, this was truly very helpful in improving this paper.

Round 2
Reviewer 2 Report
No further comments.
Author Response
Thank you for your comments
We have now addressed your suggestions regarding why we manually screened bibliographies for our review
